# Experimental Analysis of the Discharge Valve Movement of the Oil-Free Linear Compressor in the Refrigeration System

**Chengzhan Li [1],\*, Jian Sun [2],\*, Huiming Zou [3,4], Jinghui Cai [3,4] and Tingting Zhu [5]**

1   Tianjin Key Laboratory of Refrigeration Technology, Tianjin University of Commerce, Tianjin 300134, China
2   Wuhan Global Sensor Technology Co., Ltd., Guandong Street, Wuhan 430205, China
3   Key Laboratory of Space Energy Conversion Technologies, Technical Institute of Physics and Chemistry, Chinese Academy of Sciences, Beijing 100190, China
4   University of Chinese Academy of Sciences, Beijing 100049, China
5   Department of Thermal and Fluid Engineering, Faculty of Engineering Technology (ET), University of Twente, 7522 NB Enschede, The Netherlands
\*   Correspondence: lichengzhan@tjcu.edu.cn (C.L.); sunjian2106@126.com (J.S.); Tel.: +86-022-26669745 (C.L.)

**Abstract:** In a linear compressor, the valve motion significantly affects the thermodynamic efficiency and the compressor's reliability, especially in oil-free conditions. To better understand the dynamic behavior of the discharge valve, a real-time test bench was built. The piston movements and dynamic pressure in the cylinder were also observed to obtain the synchronizing characteristics among the reed valve motion, cylinder pressure, and piston motion. Observing the motion of the discharge valve visually, the discharge valve flutters due to the change in the form of the cylinder pressure, the delayed opening of the valve is caused by the inertia of the valve itself, and additional displacement fluctuations are present. This paper presents the dynamic behavior of the discharge valve under different discharge pressure/operating frequency/piston stroke/clearance length conditions. The results show that the valve flutters increase, the mean displacement of the valve increases, and the duration of the discharge increases when the discharge pressure decreases. When the operating frequency increases, the duration of the discharge decreases, while the mean displacement of the valve increases. For a high stroke or a low clearance length case, the duration of the discharge increases, while the valve flutters increase due to the pressure fluctuations in the cylinder. Through analyzing the synchronizing characteristic among the valve movements, piston movements, and cylinder pressure, it is shown that the phenomenon of the delayed opening valve is much worse for a low stroke or a high operating frequency case. In addition, the delayed closing of the valve appears for a high operating frequency case (75 Hz).

**Keywords:** oil-free linear compressor; refrigeration system; discharge valve; dynamic behavior; synchronizing characteristic

## 1. Introduction

The vapor compression refrigeration system (the VCR system) offers distinct advantages, such as a high refrigeration coefficient and a high temperature-control precision. Because of these advantages, it is widely used and accounts for approximately 80% of the refrigeration industry [1]. The VCR system can satisfy thermal comfort in buildings, storage of food or medicines, temperature and humidity regulation, and other refrigeration demands in households and industries. As the core equipment of the VCR system, the compressor mainly determines the overall performance of the system. Linear compressors have the advantages of simple structures, high efficiency, and high reliability, which have attracted the attention of many scholars and compressor manufacturers [2]. Linear compressors are still piston compressors, but they have some unique strengths compared with the conventional piston compressor. Linear compressors achieve high mechanical

efficiency because the transmission mechanism (crankshaft mechanism) between the piston and the drive motor is completely abandoned [3]. In addition, the piston stroke of a linear compressor is variable, and the cooling capacity can be simply adjusted by the stroke control. Meanwhile, flexure springs with high radial stiffness are adopted in linear compressors to achieve the gap seal between the piston and the cylinder, which means linear compressors operate without oil. The compressor operates in a resonant frequency to obtain the maximum motor efficiency due to the resonant springs. Some experimental works have evaluated the refrigeration of linear compressors in different evaporating temperatures and condensing temperatures [4–8] using different refrigerants. Furthermore, some researchers have investigated the operation characteristics of linear compressors by establishing mathematical models and experimental observations [9–12]. These works contain the operating characteristics of linear motors, the gas leakage through the gap between the piston and the cylinder, the gas force nonlinear characteristics, the piston stroke, frequency regulation, etc. Correspondingly, these characteristics aim to improve the motor efficiency and the thermodynamic efficiency.

Moreover, the valve dynamics' characteristics significantly affect the thermodynamic efficiency and the reliability of the compressors. For example, Ribas et al. [13] reported that the energy loss of the suction and discharge valve can account for nearly 50% of the thermodynamic loss in a compressor. Further, Woo et al. [14] pointed out that compressor failure usually stems from the failure of valves. For an oil-free linear compressor, the valve dynamics are particularly important. The clearance between the valve and valve plate for the oil compressors can be filled with the oil, which is beneficial to the seal of the valve. This is significantly different from the oil-free compressor. Thus, researchers should pay more attention to the reasonable valve motion for the oil-free linear compressor. The linear compressor operates at a resonant frequency that can be elevated by the increase of spring stiffness. The effect of the valve dynamics on the reliability of the compressor is deepened under the trend of high frequency and miniaturization. Therefore, the valve dynamics should focus on enhancing both the efficiency and the reliability of the linear compressor.

For a small/miniature refrigeration system, a reed valve is usually adopted in a compressor. Some researchers have investigated the valve dynamics of the linear compressor. Park et al. [15] comparatively analyzed the difference between the rigid body model and the Fluid–Structure Interaction (FSI) model of the suction valve to predict the Energy Efficiency Ratio (EER) of the compressor. The results indicated that the simple rigid body model can be used to estimate the EER of the compressor due to the low difference. However, the results from the FSI model can be close to the test results. Hwang et al. [16] held the same point by establishing the FSI model and the rigid body model of the discharge valve. Choi et al. [17] also used the FSI method to investigate the effect of the conical spring preload in a discharge valve on the dynamic behavior of valve assembly. Liang et al. [18] obtained the impact of the valve opening and closing on the cylinder pressure in a cycle by numerical analysis. Jomde et al. [19] simulated the lift of the suction and discharge valve by Finite Element Analysis (FEA), and the reed valve lifting was further tested. Reed et al. [20] measured the real-time motion of the reed valve and conducted a fatigue test of the reed valve in a linear compressor for a Joule–Thomson (J-T) cryocooler. The reed valve worked properly for 100 million fatigue cycles, which greatly exceeded the provider's quoted number of cycles (10 million cycles). Liu et al. [21] tested the dynamic behavior of a reed valve in a linear compressor for a J-T throttling cryocooler. They showed that the thermodynamic efficiency of the linear compressor was superior when the valve thickness was 0.152 mm.

In the above literature, some scholars have preliminarily tested and numerically analyzed the dynamic behaviors of the reed valve. Many mathematical models of reed valve motion have been validated by experiments. However, the experiments on valve dynamic behavior have not been systematic and specific. In particular, the dynamic behaviors of the reed valve are still scarce when the oil-free linear compressor operates under variable capacity and frequency conditions. Moreover, some reed valve tests in a linear compressor were in a nitrogen/helium atmosphere, such as the work by Liu et al. [21]. In the work

by Liu et al. [21], the linear compressor was used to drive the precooling Joule–Thomson (J-T) throttling refrigerator. The precooling Joule–Thomson (J-T) throttling refrigerator can be used to obtain the liquid helium temperature (~4 K) in a cryogenic temperature field, and the working fluid in the throttling refrigerator is helium. Therefore, the helium is compressed by the linear compressor in the precooling Joule–Thomson (J-T) throttling refrigerator. Considering the cost of experiments, i.e., the price of helium is higher than that of nitrogen, Liu et al. [21] adopted nitrogen as the working fluid since the properties of nitrogen and helium are similar to that of ideal gas. However, the refrigeration system was used to obtain the freezing or refrigerating temperature ($-10\ °C$ to $10\ °C$), which is applied in air conditioners, heat pumps, and refrigerators. The refrigerant gas, as the polyatomic gas, is a real gas whose properties are different from that of ideal gas.

Furthermore, in the work by Liu et al. [21], the viscosity of the nitrogen was about 18.1 μPa·s (the viscosity of the helium is about 20.0 μPa·s), while the viscosity of the refrigerant was about 11.1 μPa·s (taking R134a as an example). According to the rigid body model of the reed valve [15], the reed motion can be expressed by the one-dimensional vibration equation.

$$m_v \frac{d^2 x_v}{dt^2} + c_v \frac{dx_v}{dt} + k_v x_v = F_p, \tag{1}$$

where $m_v$ is the effective mass of the reed valve, $x_v$ is the displacement of the reed valve, $c_v$ is the damping coefficient, t is the time in a cycle, $k_v$ is the spring stiffness of the reed valve, and $F_p$ is the pressure difference acting on the reed valve.

By Equation (1), the damping coefficient $c_v$, which affects the motion of reed valve, is directly related to the viscosity of the working fluid. Therefore, the viscosity of the refrigerant is also different from that of the nitrogen. This is also a difference between the present work and the work by Liu et al. [21]. It can be seen that the motion of the reed valve in a vapor compression refrigeration cycle driven by a linear compressor is still rare.

Meanwhile, most tests only obtained the motion of the reed valve in a cycle, and the synchronizing characteristics with the cylinder pressure are still lacking. For an oil-free linear compressor, due to the free piston structure, the effect of the reed valve motion on the thermodynamic efficiency is still required. Consequently, a comprehensive study on the valve motion in a linear compressor is urgently needed.

In the present work, a real-time test bench was built to obtain the dynamic motion of the discharge valve in an oil-free linear compressor. This bench was capable of obtaining the cylinder pressure and piston motion. Further, the synchronizing characteristics among the reed valve motion, gas pressure in the cylinder, and piston motion are discussed. The analysis of the dynamic behaviors of the discharge valve at different discharge pressures, operating frequencies, piston strokes, and clearance lengths is presented. The study can improve the knowledge of valve dynamics and synchronization mechanisms in an oil-free linear compressor. A systematic and visual angle to the motion of the reed valve is offered, and it provides a valuable reference for optimizing the discharge valve design and improving the refrigeration efficiency of the linear compressor.

## 2. Experimental Apparatus

### 2.1. The Linear Compressor

The schematic diagram of the oil-free linear compressor is shown in Figure 1. The linear compressor contains a moving coil linear motor, a spring support structure, and a refrigerant flow structure. The moving coil linear motor consists of an inner iron, an outer iron, a copper coil, and a permanent magnet set. The high radial–axial stiffness flexure springs mainly constitute the support structure [22]. The refrigerant flow structure contains a piston, a cylinder, and a valve assembly. The copper coil directly connected to the piston winds around the titanium former, and the coil former is supported by the flexure springs. Thus, the moving part consists of the former, copper coil, and piston. The flexure springs with high radial–axial stiffness maintain the clearance seal between the piston and the cylinder, which greatly reduces the friction to achieve the oil-free operation of a linear

compressor. The suction valve and discharge valve are segregated in the linear compressor, i.e., the former is mounted on the front-end face of the piston, and the latter is mounted on the top of the cylinder. The refrigerant gas enters into the annular passage in the piston, and it is compressed by the piston and discharged out of the linear compressor. The piston is driven directly by the linear motor. When the sinusoidal voltage is imposed on the linear motor, the coil is driven by the alternating axial Ampere's force and reciprocates with the piston. With the opening and closing of the suction valve and discharge valve, the linear compressor achieves the whole process, i.e., suction, compression, discharge, and expansion. The key parameters of the linear compressor are given in Table 1. It is worth mentioning that the material of the suction valve and discharge valve was Sandvik 7C27Mo2.

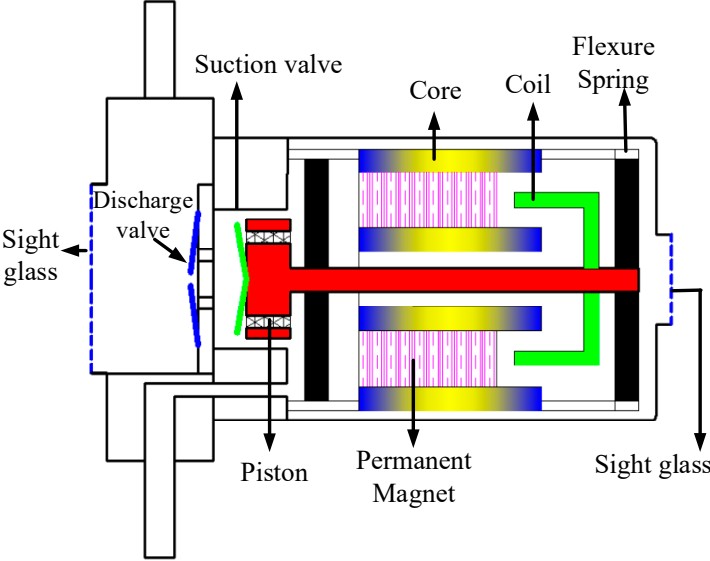

**Figure 1.** Schematic diagram of the oil-free linear compressor.

**Table 1.** The design specifications for the moving coil linear compressor.

| Items (Unit) | Value |
|---|---|
| Total mass of moving part (kg) | 0.313 |
| Resistance of motor coil (Ω) | 2.62 |
| Motor constant (N/A) | 28.5 |
| Inductance of motor coil (mH) | 3 |
| Cylinder diameter (mm) | 19 |
| Maximum stroke (mm) | 9 |
| Flux density in gas gap (T) | 0.8 |

### 2.2. The Discharge Valve Structure

The three-dimensional discharge valve structure is shown in Figure 2. The steel valve seat is placed at the top of the cylinder. In Figure 2, the straight pin (the red circle marked) in the valve plate is helpful to locate the discharge valve. The four discharge valves are attached to the valve seat. The discharge valve contains the cantilever and the arc surface. The center of the arc surface is concentric with the discharge hole. The valve stop made of PEEK material covers the discharge valve to limit the lift of the discharge valve. The other face of the valve stop is marked by the rectangular frame in Figure 2, and four grooves correspond with the four discharge valves, respectively. When the refrigerant gas pressure is higher than the discharge pressure, the discharge valve opens, and the gas enters the discharge chamber through the four discharge holes. The opening and closing of the four discharge valves are dominated by the pressure difference between the cylinder and the discharge chamber. The piston driven by the linear motor compresses the refrigerant gas to control the discharge valve.

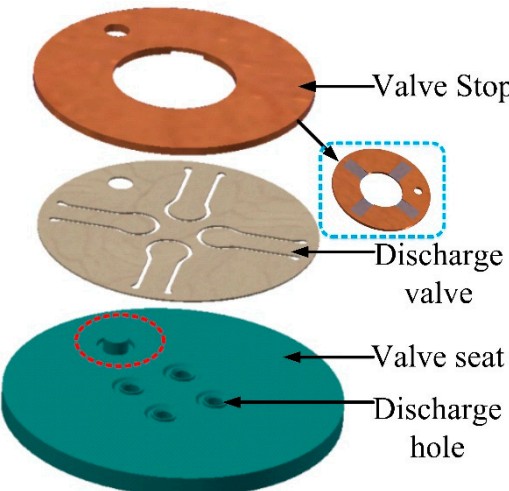

**Figure 2.** The diagram of the discharge valve structure.

### 2.3. The Test Setup

As displayed in Figure 3, the single-stage VCR system was set up to test the linear compressor. The R134a was selected as the working fluid in the experiments, because the R134a, as a current transitional hydrofluorocarbons refrigerant, has been applied in air conditioners, heat pumps, and refrigerators [23]. The VCR system contained four main components, i.e., an evaporator, a linear compressor, a condenser, and a throttle valve. Some auxiliary components were also part of the refrigeration system, such as the dry filter and the flowmeter. In this system, the low-pressure refrigerant gas flows from the evaporator to the linear compressor. In the compressor, the refrigerant gas is compressed into high-pressure and high-temperature gas, which is discharged into the condenser. In the condenser, the refrigerant gas releases heat to the surrounding environment and turns into the refrigerant liquid. The refrigerant liquid flows through the throttle valve. The mixture of refrigerant liquid and vapor enters the evaporator. The mass flow meter is located between the dry filter and the throttle valve to obtain the mass flow rate of refrigerant liquid. The dry filter is used to dry and filter the refrigerant. Two pressure sensors were mounted in the inlet and outlet of the linear compressor to measure the suction pressure and discharge pressure of the compressor. Six thermal sensors were installed at the inlet and outlet of the linear compressor, condenser, throttle valve, and evaporator. As indicated in the Introduction, the dynamic pressure in the cylinder was also recorded by a miniature dynamic high-frequency pressure sensor in real time to obtain the synchronizing characteristics between the cylinder pressure and the valve movements. Considering the compressor inner structure and the dimensions of the dynamic pressure sensor and the cylinder, the dynamic pressure sensor was installed on the side of the cylinder, and the installation construction is shown in Figure 4. The miniature pressure sensor was embedded into the threaded hole (M3.5 × 0.6) positioned at the side of the cylinder wall. The side thread hole (M3.5 × 0.6) was connected with the clearance hole (Φ1.5), which directly connected to the inside of the cylinder. By the above-mounted structure, the miniature pressure sensor could measure the real-time pressure of the cylinder. In order to obtain the movement of the piston, a visual window was mounted at the rear of the linear compressor. The laser displacement sensor was set up to measure the real-time displacement of the piston. The measurement method used was the laser triangulation method. The laser beam was subjected to the end of the piston rod, and diffuse reflection occurred when reaching the end surface of the piston rod. Part of the laser beam was received by the detector. The displacement was calculated by the laser displacement sensor and shown in the display window. The displacement of the valve was measured by the same method. The difference was that the visual window was located in the discharge chamber of the front end of the linear compressor. The laser beam was subjected to the center of the arc surface of the discharge valve, and then the

displacement/lift could be recorded. The visual window contained two quartz glasses (6 mm thickness), a seal structure made of Teflon, and a front/rear cover plate. In addition, a power meter was applied to record the electrical parameters of the linear compressor (voltage, current, power factor, and frequency). A power supply was used to provide the driving voltage and drive the linear compressor with suitable operation parameters, such as the variable piston displacement and the drive frequency.

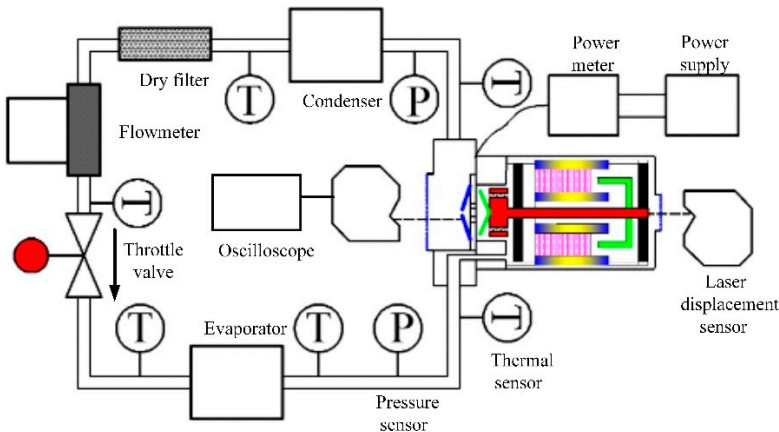

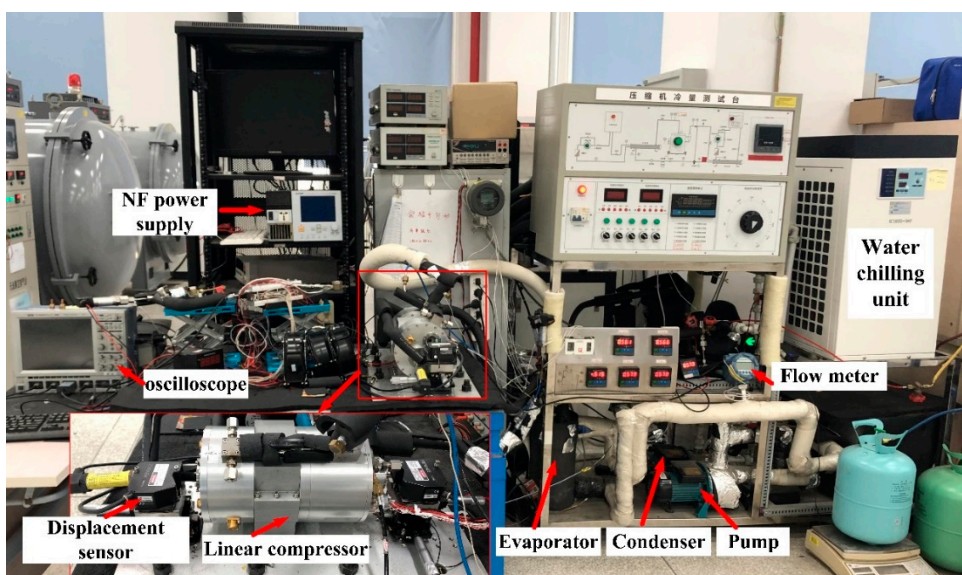

**Figure 3.** The schematic and real picture of the VCR system.

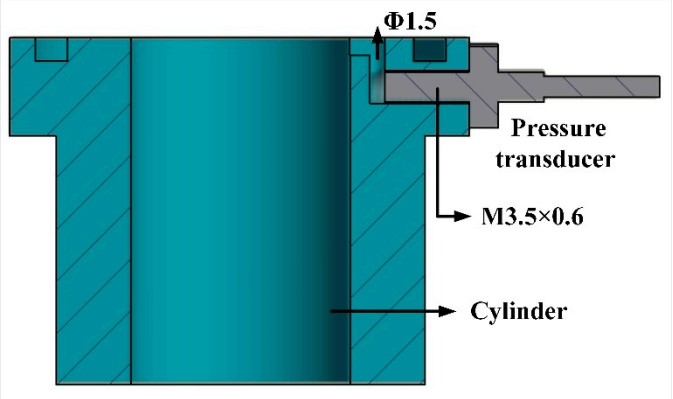

**Figure 4.** The schematic of the dynamic pressure sensor installation.

### 2.4. The Data Acquisition System

The flowchart of the real-time data acquisition system is exhibited in Figure 5. For this acquisition system, there was a high-speed data acquisition card and a low-speed data acquisition card. The high-speed data acquisition card, which has 8 signal input channels, was used to record the cylinder pressure, piston displacement, and valve lift. The low-speed data acquisition card was used to obtain the temperature, the suction and discharge pressures, and the mass flow rate. The sample rate of the high-speed data acquisition cards was 250,000 samples per second. For the dynamic pressure sensor, the voltage signal was first amplified by the signal amplifier and then collected by the high-speed acquisition cards. The real-time curve of the piston displacement, valve lift, and cylinder pressure were displayed by the oscilloscope. The oscilloscope stored the data of the dynamic signals. The static parameters (the temperature, the suction and discharge pressures, and the mass flow rate) were recorded once per second. All the acquisition signals from the two cards were stored in a computer via the Labview Software to analyze and process the data. The displacement curves of the discharge valve were recorded under thermal steady-state conditions. For each measurement condition, the thermal steady-state condition for the refrigeration system was achieved when the variation in the measured temperature was below 0.5 K within 10 min [24].

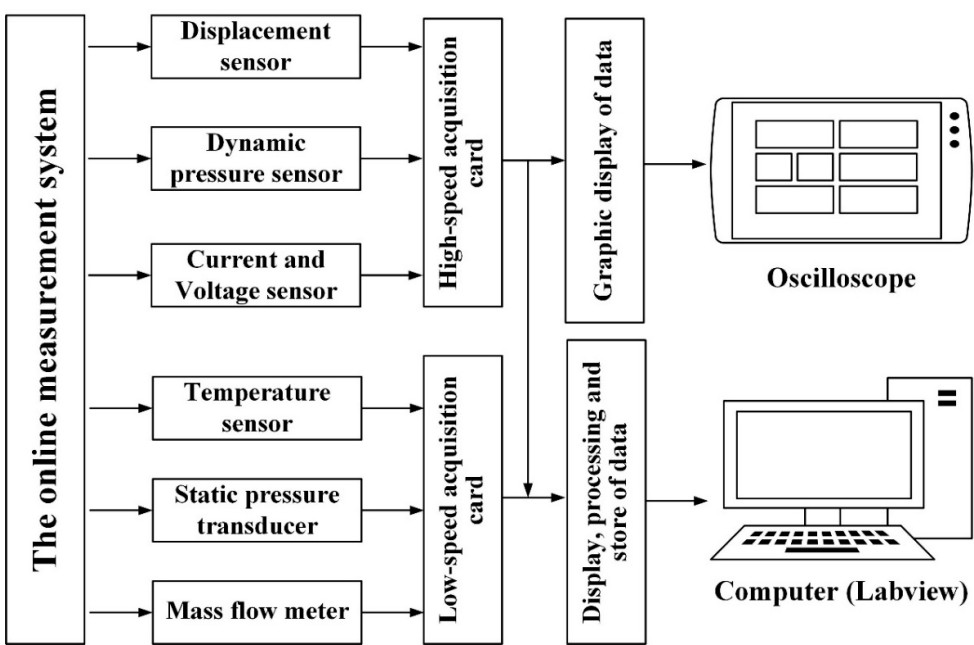

**Figure 5.** The flowchart of the data acquisition system.

## 3. Results and Discussion

### 3.1. The Movements of the Discharge Valve under Different Discharge Pressures

Figure 6 shows the displacement curves of the discharge valve under different discharge pressures. The stroke of the piston was 8 mm, the suction pressure was 0.35 MPa, and the operation frequency was 50 Hz. By adjusting the throttle valve and input voltage of the compressor, the discharge pressure was set as 0.75 MPa, 0.85 MPa, and 0.94 MPa, respectively. The displacement curves of the discharge valve had some similar features under different discharge pressures. The valve rapidly opened when the cylinder pressure was maximum. Then, the displacement of the discharge valve reached the maximum, and the valve slightly fell. The valve rebounded again, and the displacement of the valve was less than the displacement of the valve when it opened the first time. The discharge valve fell back onto the valve seat. In the process of falling back, the valve appeared to rebound due to hitting the valve seat and then gradually fluctuated until clinging to the valve seat. For the lower discharge pressure, the closing time of the valve was earlier, the rebound

displacement was higher, and the duration of the valve opening was longer. This means that the flow area of the refrigerant gas through the discharge valve decreased, and thus the pressure loss in the discharge process increased with the increase in the discharge pressure. As shown in the partially enlarged drawing in Figure 6, the valve appeared to flutter. This can be explained by the change in the form of pressure. When the valve was not opened, the static pressure in the cylinder was total pressure. As the valve opened, the refrigerant gas began to flow. Part of the total pressure was transformed into the dynamic pressure, and thus the static pressure decreased (the sum of the static pressure and the dynamic pressure is the total pressure). When the valve opened, the static pressure difference on the discharge valve was higher than the spring force from the valve itself. With the static pressure in the cylinder, the static pressure difference on the valve decreased, and thus the valve could not maintain the initial position. Therefore, the displacement of the valve decreased. However, with the refrigerant gas compressed by the piston, the static pressure in the cylinder increased, and thus the valve displacement increased again. In addition, the discharge pressure had an effect on the flutter of the valve in the discharge process. In the lower discharge pressure, the number of flutters increased, which was also due to the pressure fluctuation in the cylinder. According to the reference [25], a higher discharge pressure leads to a larger force on the discharge valve and a faster speed, which shortens the discharge duration with the increase in the discharge pressure.

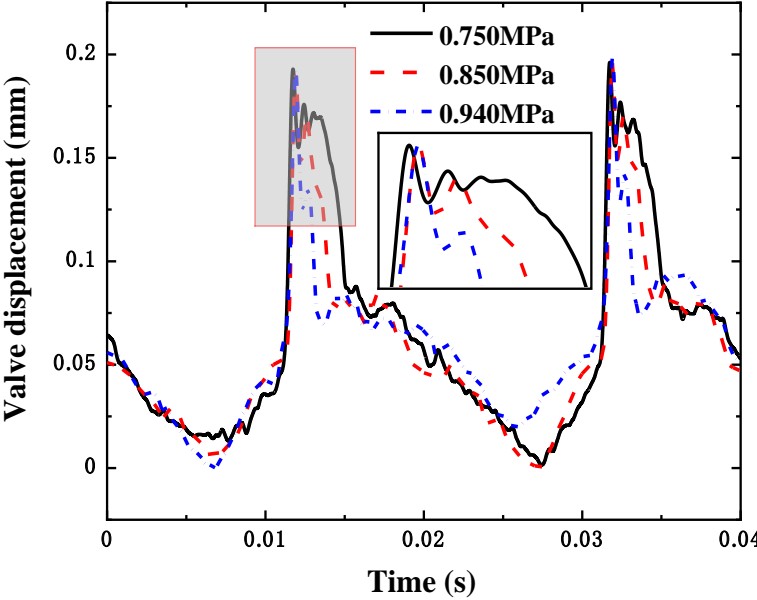

**Figure 6.** Displacement curves of the discharge valve under different pressures.

### 3.2. The Movements of the Discharge Valve under Different Frequencies

By adjusting the power supply, the operation frequencies of the linear compressor were set to 35 Hz, 50 Hz, 62.5 Hz, and 75 Hz, and the piston stroke remained constant at 8 mm. By adjusting the throttle valve, the suction and discharge pressures were 0.35 MPa and 0.87 MPa, respectively. Figure 7 shows the displacement curves of the discharge valve at different operating frequencies. With the increase in the operation frequency, the duration of the discharge decreased, as expected. The mean displacement of the valve increased with the operation frequency, which was due to the higher static pressure in the cylinder for the higher frequency (as shown in Figure 11). It can be seen from Figure 7 that the oscillation period of the valve movement was almost identical at different operation frequencies, which demonstrates that the flutter of the valve was independent of the operating frequency. Similar valve behavior was also observed in the references [26,27]. This is because the parameters and the natural frequency of the discharge valve did not

change, and the oscillation period of the valve movement is related to the natural frequency of the valve [27].

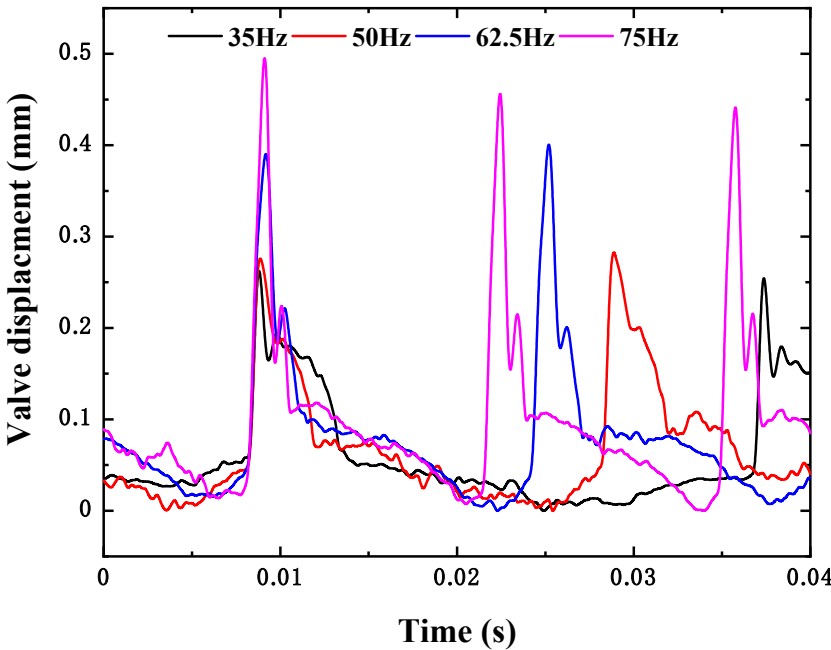

**Figure 7.** Displacement curves of the discharge valve under different frequencies.

*3.3. The Movements of the Discharge Valve under Different Strokes and Different Clearance Lengths*

Due to the free piston structure, the piston stroke of the linear compressor can be adjusted by changing the input voltage. The dynamic behavior of the discharge valve under different strokes and different clearance lengths is discussed in this part.

The operation frequency was kept constant at 50 Hz. The suction and discharge pressures were set to 0.35 MPa and 0.81 MPa, respectively. The piston stroke was adjusted to 7 mm, 7.5 mm, 8 mm, 8.5 mm, and 9 mm by regulating the input voltage. The displacement curves of the discharge valve under different strokes are displayed in Figure 8. The mean displacement of the discharge valve, the time section enclosed by the displacement curve, and the duration of discharge increased with the piston stroke, which means that the clearance length reduced, but the volume efficiency and the discharge duration of a cycle increased for a higher piston stroke. The enhancement of the valve mean displacement can be helpful to reduce the pressure loss in the process of discharge. The number of flutters increased against the piston stroke. For a longer discharge time, the fluctuation in the static pressure in the cylinder increased, which induced more flutters. In addition, the number of flutters increased the pressure loss.

The piston stroke was kept constant at 8 mm, and the suction pressure and discharge pressure were 0.35 MPa and 0.81 MPa, respectively. The clearance lengths were adjusted to 1 mm, 2 mm, and 3 mm by modulating the input voltage. Figure 9 shows the displacement curves of the discharge valve under different clearance lengths. The duration of the discharge increased with the clearance length. The lower the clearance length was, the longer the discharge time accounting for a cycle became. The time section enclosed by the displacement curves of the valve increased with the decrease in the clearance length, which also contributed to reducing the pressure loss, and it was identical to the different stroke cases. For a constant stroke, the clearance length had little effect on the mean displacement of the valve, and the number of flutters increased with the decrease in the clearance length. Similar to the cases at different strokes, the number of flutters was affected by the static pressure fluctuation in the cylinder.

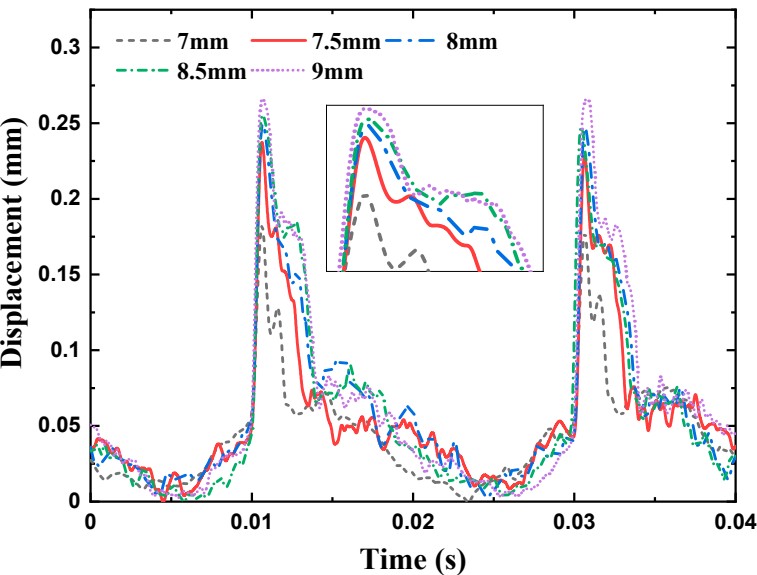

**Figure 8.** Displacement curves of the discharge valve under different strokes.

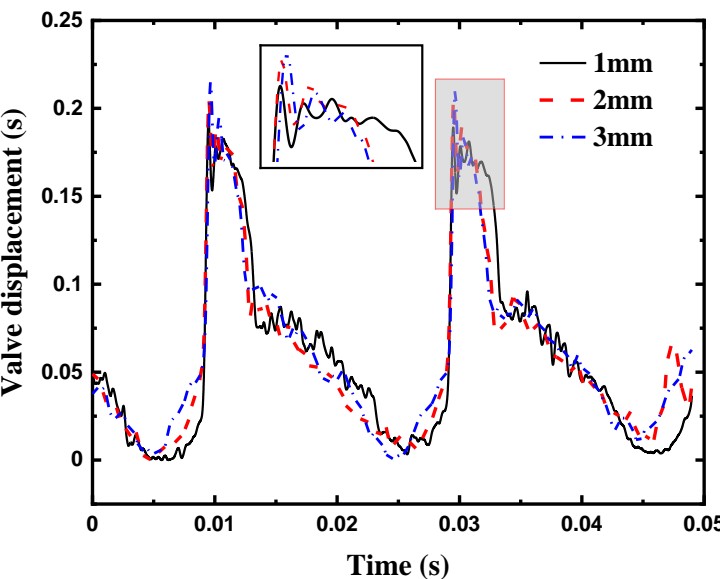

**Figure 9.** Displacement curves of the discharge valve under different clearance lengths.

### 3.4. The Discharge Valve Movements with the p-t Diagram inside the Cylinder and Piston Motion

The piston stroke and operation frequency of the linear compressor can be regulated, which is different from the conventional piston compressor. Therefore, this part gives the synchronizing characteristics between the piston movements, valve movements, and cylinder dynamic pressure.

Figure 10 shows the displacement curves of the discharge valve and piston along with the pressure–time diagram in the cylinder at different strokes (7 mm, 8 mm, and 9 mm). When the cylinder pressure was higher than the discharge pressure, the valve began to open. However, the time of the valve opening was slightly delayed, which is due to the inertia of the discharge valve. With the decrease in the piston stroke, the delay of the valve opening was worse, which indicates that the pressure difference between the cylinder and the discharge chamber is lower for a smaller stroke. When the cylinder pressure was lower than the discharge pressure, the piston was at Top Dead Center, and the discharge valve closed. Overall, the closing time of the discharge valve was almost identical to the time when the piston was at Top Dead Center. After the piston was at Top Dead Center,

additional displacement fluctuation was generated, which induced the backflow of the refrigerant gas from the discharge chamber to the cylinder. Consequently, the backflow of the high-pressure gas contributed to the reduction in the volume efficiency. When valve assemblies are designed, the roughness of the valve plate should be improved to reduce the unevenness of the valve plate surface between the discharge valve and the valve seat.

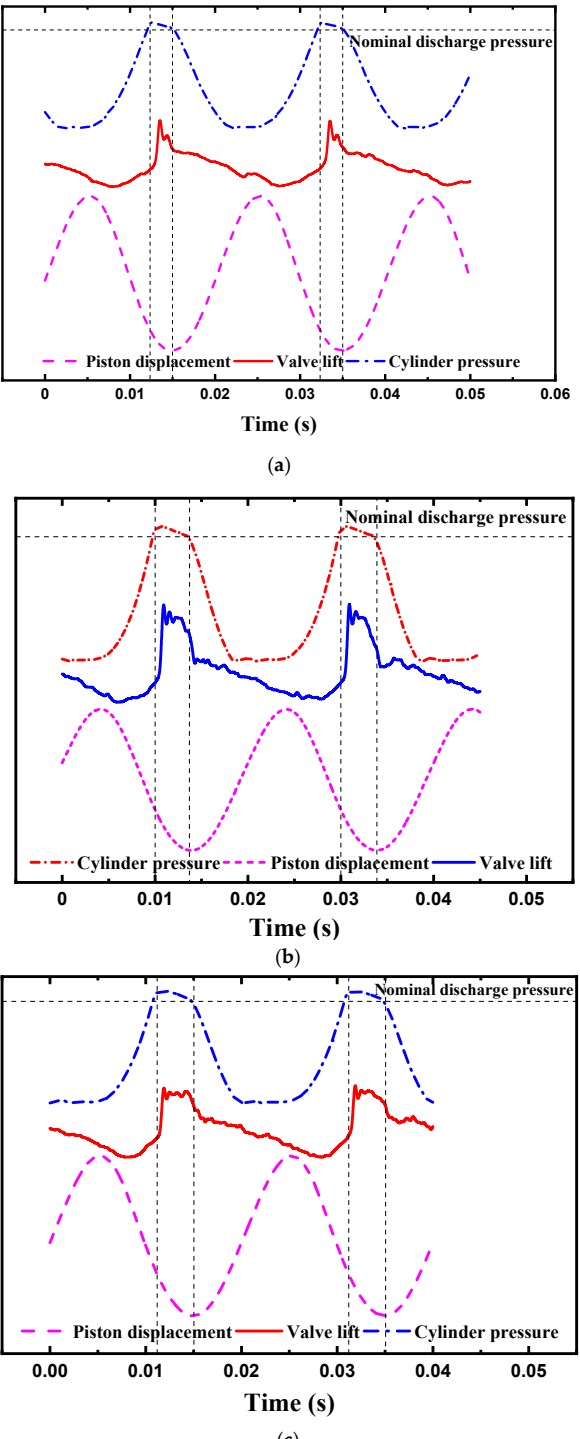

**Figure 10.** The relationship between the valve displacement, cylinder pressure, and the piston motion at different strokes (valve displacement also called valve lift). (**a**) Stroke = 7 mm. (**b**) Stroke = 8 mm. (**c**) Stroke = 9 mm.

Figure 11 shows the displacement curves of the discharge valve and piston along with the pressure–time diagram in the cylinder under different frequencies (35 Hz, 50 Hz, 62.5 Hz, and 75 Hz). It can be seen that the delay of the opening time was worse with the operation frequency. This is mainly due to the inertia of the valve. The inertia force is proportional to the square of the frequency [15]. Meanwhile, the slight delay of the valve closing was present when the piston was at the Top Dead Center, which also induced the backflow of the gas. Therefore, the opening and closing time of the valve was delayed at a low operation frequency. For the design of the discharge valve, the mass of the valve should be reduced to avoid the delay phenomenon.

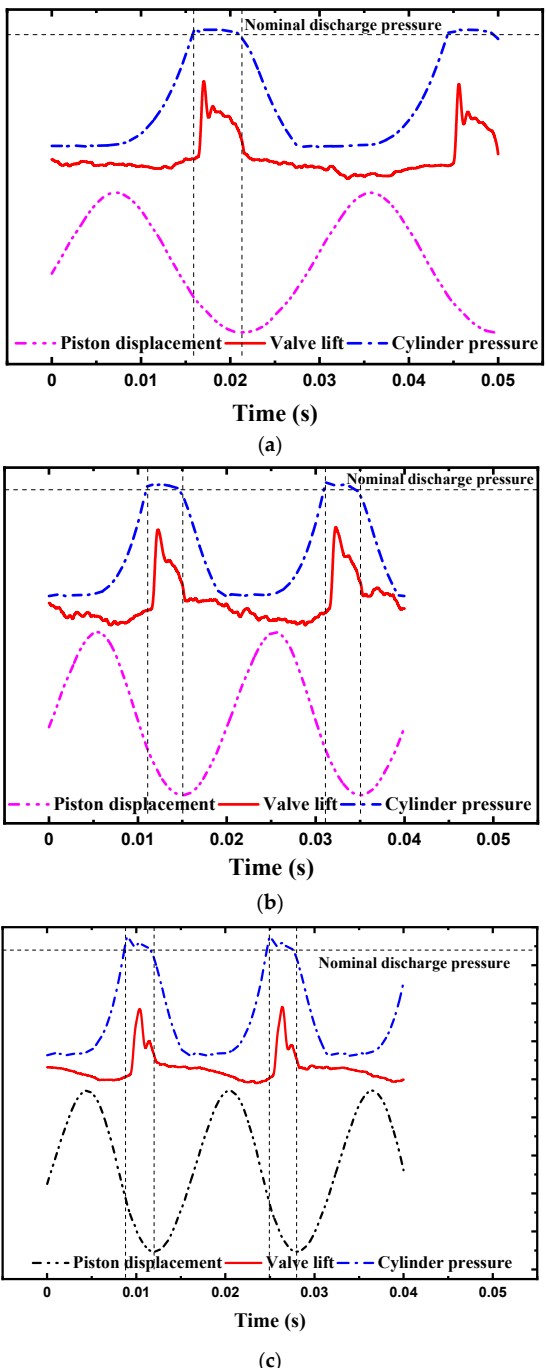

**Figure 11.** *Cont.*

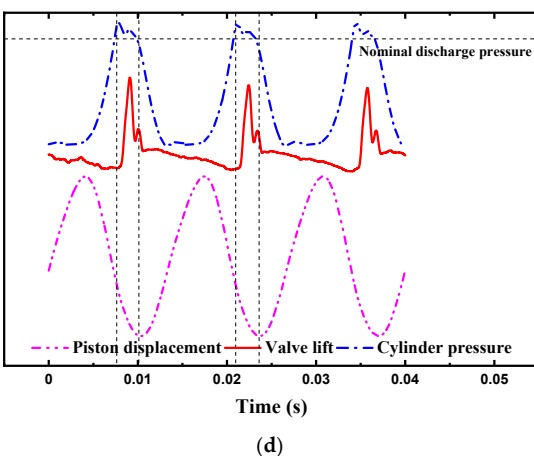

(**d**)

**Figure 11.** The relationship between the valve displacement, cylinder pressure, and piston motion under different frequencies. (**a**) Frequency = 35 Hz. (**b**) Frequency = 50 Hz. (**c**) Frequency = 62.5 Hz. (**d**) Frequency = 75 Hz.

## 4. Conclusions

In the present work, the dynamic movements of the discharge valve in an oil-free refrigeration linear compressor were observed by a real-time test bench. Meanwhile, the piston movements and dynamic pressure in the cylinder were also measured. Furthermore, the dynamic behavior of the discharge valve under different conditions was discussed. The synchronizing characteristics between the reed valve motion, cylinder pressure, and piston movements were also analyzed.

1. By observing the time-domain curves of the discharge valve displacement, the dynamic behavior is visually understood. The discharge valve flutters due to the change in the form of pressure in the cylinder, which changes from static pressure to dynamic pressure. The delayed opening of the valve is caused by the valve inertia. Additional displacement fluctuations are present, which is due to the unevenness of the valve plate surface between the valve and the valve seat.
2. With the decrease in the discharge pressure, the valve flutters increase due to the pressure fluctuations in the cylinder, the mean displacement of the valve increases due to the high static pressure difference between the cylinder and the discharge chamber, and the duration of the discharge increases due to the low speed of the valve.
3. With the increase in the operation frequency, the duration of the discharge decreases, but the mean displacement of the valve increases due to the higher static pressure in the cylinder for the higher frequency. The oscillation period of the valve movement is almost identical under different operation frequencies; this is because the parameters of the discharge valve do not change.
4. For a high stroke and a low clearance length, the duration of the discharge increases due to the increase in the volume efficiency and the increase in the duration of the discharge for a cycle, and the valve flutters increase due to the pressure fluctuations in the cylinder.
5. In terms of the relationship between the valve movements, piston movements, and cylinder pressure, the delayed closing of the valve is little affected by the piston stroke, while the delayed opening of the valve is evidently affected at a low stroke (7 mm) due to the low pressure difference between the cylinder and the discharge chamber, and the delayed opening of the valve is more evident with the increase in the operating frequency. The delayed closing of the valve occurs at a high frequency (75 Hz).

**Author Contributions:** C.L.: Conceptualization, Methodology, Writing—Original draft preparation; Reviewing and Editing; J.S.: Data curation, Experiment, Reviewing and Editing. H.Z.: Investigation, Supervision; J.C.: Methodology, Supervision; T.Z.: Text Editing and Validation. All authors have read and agreed to the published version of the manuscript.

**Funding:** This work was financially supported by the National Natural Science Foundation of China (Grant NO. 51976229 and NO. 52276023). The authors are grateful for the support of the sponsors.

**Institutional Review Board Statement:** Not applicable.

**Informed Consent Statement:** Not applicable.

**Data Availability Statement:** Not applicable.

**Conflicts of Interest:** The authors declare no conflict of interest.

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
