# Peer review of "Experimental Analysis of the Discharge Valve Movement of the Oil-Free Linear Compressor in the Refrigeration System"

_sustainability, doi:10.3390/su15075853_

Round 1

Reviewer 1 Report

Dear Authors,

In your manuscript interesting experimental results of the experimental analysis of the discharge valve movement of the oil-free linear compressor in the refrigeration system are provided.

The main issue is the high level of similarity (even the same figures) with the paper already published in Applied Thermal Engineering 2022 "Experimental investigation of the discharge valve dynamics in an oil-free linear compressor for Joule-Thomson throttling refrigerator (Yuanli Liu, Jian Sun, Yuqiang Xun, Zhijie Huang, Jinghui Cai, Chengzhan Li).

You should provide clear explanation of the differences of these two manuscripts in order to convince the readers of the necessity for one more published paper on these topics.

The language should be also improved since there are several basic-level errors noted (e.g. Page 2 "their manifested", "the Nitrogen and helium", etc.) and parts of the text sound artificial.

Unify the references enlisted in References sources according to the common practice of this journal (e.g. in [24] "[J]" left, etc.). 

Reviewer 2 Report

I have read the article titled " Experimental Analysis of the Discharge Valve Movement of the Oil-free Linear Compressor in the Refrigeration System ". This article intends to observe the relief valve's dynamic movements with an oil-free refrigeration linear compressor and a real-time test bench. My comments are below.

- 21 references were used in the introduction part of the article. This is a sufficient number. However, no references are used throughout the second part. This is a severe problem for an article.

- "From the relationship among..." part is used approximately similarly in the abstract and results part of the article.

The above problems indicate that the article was not seriously planned.

Reviewer 3 Report

Comments to the Authors

sustainability-2253736-peer-review-v1: " Experimental Analysis of the Discharge Valve Movement of the Oil-free Linear Compressor in the Refrigeration System", Chengzhan Li et al.  

The authors have built real-time test bench to study of the dynamic behavior of the discharge valve, the piston movements and dynamic pressure in the cylinder

The review paper will be improved if the following problems are solved.

1.      Grammar

Grammar should be checked throughout the manuscript. For example, “And the piston movements and dynamic pressure in the cylinder are also observed.” On the 3-4 lines in the abstract. “And” should be deleted.

2.      Experiment

As the focus of this manuscript is experiment. The real picture of the “real-time test bench” should be shown for better understanding.

3.      Discussion of results

In figures 8 and 9, three parameters are selected for each figure. But the tendency is the same or no peak point and turning point. So the authors should explain two questions : 1) whether the selected 3 parameters are typical or not; 2) selected 3 parameters are enough or not for research.

Round 2

Reviewer 1 Report

Dear Authors,

It seems that you responded to the issues raised during the previous reviewing stage. Hence, I recommend this version of your manuscript for publishing in this journal.

Reviewer 2 Report

Accept

Reviewer 3 Report

As the authors have revised the manuscript properly according to the comments, it could be accepted for publication.